

# Hi-MC: a novel method for high-throughput mitochondrial haplogroup classification

Sandra Smieszek[1,2], Sabrina L. Mitchell[3], Eric H. Farber-Eger[4], Olivia J. Veatch[5], Nicholas R. Wheeler[1,2], Robert J. Goodloe[6], Quinn S. Wells[7,8], Deborah G. Murdock[9] and Dana C. Crawford[1,2]

[1] Population and Quantitative Health Sciences, Case Western Reserve University, Cleveland, OH, USA
[2] Institute for Computational Biology, Case Western Reserve University, Cleveland, OH, USA
[3] Vanderbilt Eye Institute and Department of Ophthalmology & Visual Sciences, Vanderbilt University Medical Center, Nashville, TN, USA
[4] Vanderbilt Institute for Clinical and Translational Research, Vanderbilt University Medical Center, Nashville, TN, USA
[5] Department of Neurology, Vanderbilt University Medical Center, Nashville, TN, USA
[6] Center for Human Genetics Research, Vanderbilt University, Nashville, TN, USA
[7] Department of Medicine, Vanderbilt University Medical Center, Nashville, TN, USA
[8] Department of Pharmacology, Vanderbilt University, Nashville, TN, USA
[9] Center for Mitochondrial and Epigenomic Medicine, Children's Hospital of Philadelphia, Philadelphia, PA, USA

Corresponding author
Dana C. Crawford,
dana.crawford@case.edu

## ABSTRACT

Effective approaches for assessing mitochondrial DNA (mtDNA) variation are important to multiple scientific disciplines. Mitochondrial haplogroups characterize branch points in the phylogeny of mtDNA. Several tools exist for mitochondrial haplogroup classification. However, most require full or partial mtDNA sequence which is often cost prohibitive for studies with large sample sizes. The purpose of this study was to develop Hi-MC, a high-throughput method for mitochondrial haplogroup classification that is cost effective and applicable to large sample sizes making mitochondrial analysis more accessible in genetic studies. Using rigorous selection criteria, we defined and validated a custom panel of mtDNA single nucleotide polymorphisms that allows for accurate classification of European, African, and Native American mitochondrial haplogroups at broad resolution with minimal genotyping and cost. We demonstrate that Hi-MC performs well in samples of European, African, and Native American ancestries, and that Hi-MC performs comparably to a commonly used classifier. Implementation as a software package in R enables users to download and run the program locally, grants greater flexibility in the number of samples that can be run, and allows for easy expansion in future revisions. Hi-MC is available in the CRAN repository and the source code is freely available at https://github.com/vserch/himc.

## INTRODUCTION

Human mitochondrial DNA (mtDNA) consists of a double-stranded, circular chromosome that spans 16,529 base pairs and encodes 22 transfer RNAs, two ribosomal

RNAs, and 13 proteins that are part of the oxidative phosphorylation enzyme complexes. Compared with nuclear DNA, unique characteristics of mtDNA include uniparental (i.e., matrilineal) inheritance, lack of recombination, high copy number, and a high mutation rate. These characteristics make mtDNA a powerful tool for investigations in multiple disciplines including population and medical genetics, molecular anthropology, and forensics (*Chaitanya et al., 2014*). Strong evidence exists supporting the involvement of mtDNA variation in human disease phenotypes, underscoring the importance of integrating the mitochondrial genome into genetic association studies. Evidence includes the association of mtDNA single nucleotide polymorphisms (SNPs) and mitochondrial haplogroups with a number of phenotypes encompassing cancer, neurologic, ocular, cardiovascular, and metabolic traits (*Fetterman et al., 2013*; *Hudson et al., 2014*; *Shamnamole et al., 2013*; *Mitchell et al., 2014a*; *Van der Walt et al., 2004*; *Wallace, 2013*).

Mitochondrial haplogroups are collections of similar combinations of mtDNA SNPs inherited from a common ancestor. These haplogroups are formed via the sequential accumulation of mutations through the maternal lineage. As a result of population migration, distinct mitochondrial haplogroups are associated with different continental ancestries including African, European, Native American, Asian, and Oceanic (*Forster, 2004*; *Maca-Meyer et al., 2001*; *Wallace, 2013*), allowing for accurate classification of maternal genetic ancestry in large datasets using a small subset of mitochondrial markers.

Currently, several methods are available for mitochondrial haplogroup classification including Haplogrep2, HaploFind, MitoTool, HmtDB, MToolBox, and Phy-mer (*Calabrese et al., 2014*; *Fan & Yao, 2011*, *2013*; *Kloss-Brandstätter et al., 2011*; *Navarro-Gomez et al., 2015*; *Rubino et al., 2012*; *Vianello et al., 2013*; *Weissensteiner et al., 2016*). While these methods are powerful tools for mtDNA sequence analysis, including classification of mitochondrial haplogroups, most require full or partial mtDNA sequence, and some are limited in the number of samples that can be processed at once. To address limitations of existing methods we developed a high-throughput method for automated mitochondrial haplogroup classification that can accommodate large sample sizes with SNP data recorded in the widely used pedigree (PED/MAP) file format.

Using a custom panel of mitochondrial SNPs, we constructed a reduced mitochondrial phylogenetic tree, and developed an algorithm (Hi-MC) for broad classification of European, African, and Native American mitochondrial haplogroups. We then genotyped and assigned mitochondrial haplogroups to DNA samples used by the International HapMap Project (CEU, YRI, CHB/JPT, and MXL) (*International HapMap Consortium, 2003*; *International HapMap Consortium, 2005*, *International HapMap 3 Consortium, 2010*). To evaluate the performance of the algorithm, we compared these Hi-MC mitochondrial haplogroup classifications in DNA samples used by the HapMap Project with haplogroup classifications previously reported by HapMap. We also compared these haplogroup classifications with classifications assigned for the same DNA samples with genotypes from the custom SNP panel using Haplogrep2, the most widely used web-based application for mitochondrial haplogroup classification. As a final comparison, we assigned haplogroups via Haplogrep2 to the subset of overlapping DNA samples sequenced by the 1,000 Genomes Project (*The 1,000 Genomes Project Consortium, 2010*, *2012*). In general, Hi-MC assignments

were highly concordant with both HapMap-reported and Haplogrep2 assignments for CEU. Hi-MC haplogroup assignments were also highly concordant with HapMap-reported and Haplogrep2 YRI assignments with the latter using 1,000 Genomes data. When compared with HapMap-reported assignments, neither Hi-MC nor Haplogrep2 were highly concordant for CHB/JPT samples. However, Hi-MC and Haplogrep2 classifications were highly concordant with each other suggesting HapMap-reported assignments for this population are incorrect. Overall, these data suggest that Hi-MC and its associated custom SNP panel perform well compared with a popular classifier. Although it may not yet offer the same resolution as full sequence data, Hi-MC and its associated custom SNP list provide a user-friendly method for high-throughput classification provided in an R software package that can be easily expanded in future revisions to capture additional mitochondrial haplogroups.

## MATERIALS AND METHODS

### Algorithm

The algorithm input is a list of mitochondrial SNP genotypes for each individual DNA sample, and the output is haplogroup classification. The revised Cambridge reference sequence is used to specify SNP positions. Phylotree, a comprehensive phylogenetic tree of human mtDNA variation displaying relationships between mitochondrial haplogroups (*Van Oven & Kayser, 2009*), was used as a reference to create a reduced tree of 46 common haplogroups as presented in *Mitchell et al. (2014b)*. This reduced classification tree was converted into a node-based tree structure. Each haplogroup node has a list of associated SNPs, a parent node, and zero or more child nodes. The SNPs associated with a node define which SNP genotypes a subject must possess to belong to the corresponding haplogroup. Classification into a haplogroup also requires a subject to recursively meet the definition for the parent haplogroup. Haplogroups that require the reversion to the ancestral genotype (e.g., 10398A–10398G) are accommodated by adding a second hierarchy of required SNP genotypes.

The algorithm determines the appropriate haplogroup in a two-step process (Fig. 1). In the first step, the algorithm passes mitochondrial SNP genotype data for each subject into the root node of the tree. The algorithm checks the list of SNP genotypes against those required by the root node. If the genotypes meet the criteria for the parent node, this haplogroup is added to an accumulator. The algorithm then passes the list of SNP genotypes to each of the child nodes connected to that parent node until the tree is exhausted. Next, the algorithm ranks the list of haplogroups in the accumulator according to their distance from the root node. Any haplogroup with a path length less than that of the haplogroup with the longest path length is dropped. The remaining haplogroups, along with their path from the root node to the end node, are returned as a result.

### Implementation

The algorithm is implemented as a package in R (*R Core Team, 2013*, https://github.com/vserch/himc) and is available in the CRAN repository. Data input is standard PED/MAP formatted files. The output is an R dataframe object that includes subject IDs with a corresponding haplogroup classification and the path through the tree from

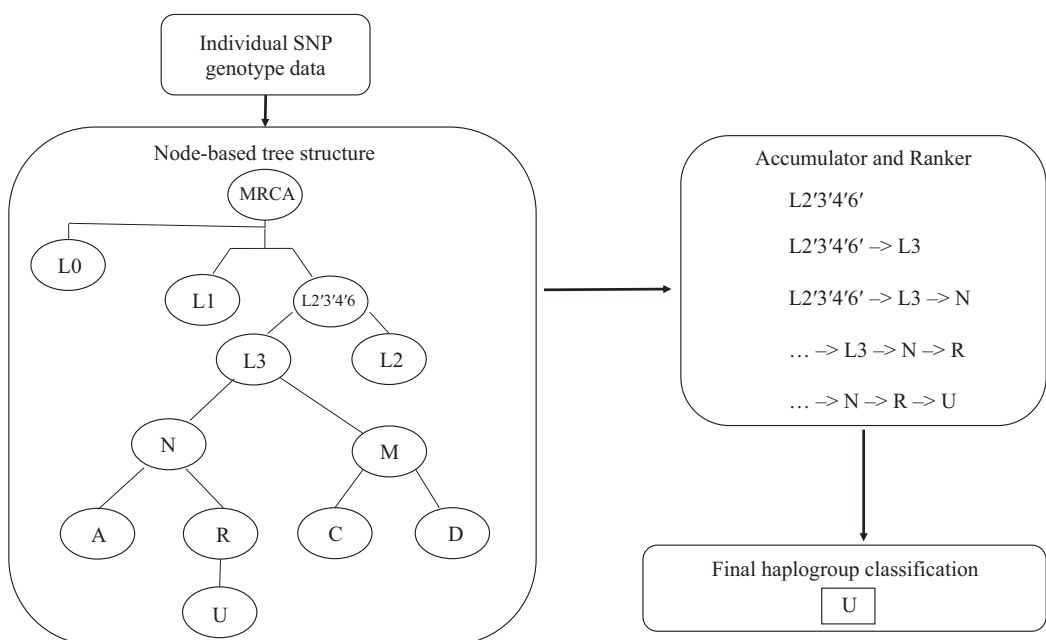

**Figure 1 Hi-MC algorithm structure.** Input for the algorithm is a list of sample IDs and corresponding SNP genotype data in pedigree (PED/MAP) format. These genotypes are recursively analyzed through a node-based tree structure. Each successive genotype classification is passed on to the Accumulator. They are then ranked according to specificity (longer path through the tree -> more SNPs checked -> more specific), with the most specific haplogroup as the final output. MRCA, most recent common ancestor.

root node to final classification. The output can easily be exported directly to a CSV file or text file. For further details on use of the Hi-MC package in R visit http://www. icompbio.net/resources/software-and-downloads/.

## Mitochondrial SNP selection

The SNPs were selected for broad classification of European, African, and Native American mitochondrial haplogroup lineages as previously described (*Mitchell et al., 2014b*). Briefly, SNPs were chosen using Phylotree (*Van Oven & Kayser, 2009*) and an extensive literature search for prior studies related to mitochondrial haplogroup classification (*Herrnstadt et al., 2002*; *Paneto et al., 2011*; *Poole et al., 2010*; *Van der Walt et al., 2003*). Preference was given to those SNPs that appear only once in Phylotree since such SNPs are specific to a single haplogroup. A total of 63 SNPs were selected, the majority of which are located in the coding region of the mitochondrial genome. Three Sequenom genotyping assay pools including all of these SNPs were designed using the MassARRAY software (*Mitchell et al., 2014b*). As described in *Mitchell et al. (2014b)*, the custom SNP panel was genotyped in 19,608 participants from the National Health and Nutrition Examination Surveys (NHANES) accessed by the Epidemiologic Architecture for Genes Linked to Environment (EAGLE) (*Crawford et al., 2015*), a study site of the Population Architecture using Genomics and Epidemiology (PAGE) I study (*Matise et al., 2011*). The Vanderbilt University Institutional Review Board determined that EAGLE was "non-human" subjects research.

## Application of Hi-MC

To evaluate the performance of Hi-MC for mitochondrial haplogroup classification we genotyped the custom SNP panel in, and applied the algorithm to, HapMap Phase I and Phase III samples. We selected HapMap DNA samples for the present study as HapMap DNA samples were the preferred reference DNA samples for individual study sites, including this study as part of the larger PAGE I study (*Matise et al., 2011*). The populations from HapMap Phase I included: individuals of Northern and Western European ancestry from the Centre d'Etude du Polymorphisme Humain samples collected in Utah, USA (CEU, $n = 90$), Yoruba from Ibadan, Nigeria (YRI, $n = 90$), Japanese in Tokyo, Japan (JPT, $n = 45$), and Han Chinese in Beijing, China (CHB, $n = 45$). The HapMap Phase III samples used in this study included only those of Mexican ancestry from Los Angeles, California (MXL, $n = 90$). The International HapMap Consortium reported mitochondrial haplogroup classifications for the CEU, YRI, CHB, and JPT Phase I HapMap samples (available at ftp://ftp.ncbi.nlm.nih.gov/hapmap/mtDNA_and_chrY_haplogroups/) (*International HapMap Consortium, 2005*); however, mitochondrial haplogroup classifications for the Phase III MXL samples have not been previously reported by HapMap.

We genotyped the custom SNP panel in the CEU, YRI, and CHB/JPT Phase I HapMap samples and in the MXL samples from Phase III. Briefly, aliquots of DNA from HapMap CEU, YRI, CHB/JPT, and MXL samples were obtained from the Coriell Institute for Medical Research. Cell line provenance information from which these DNA samples were extracted can be found within the NHGRI International HapMap Collection at https://www.coriell.org/1/NHGRI. SNPs were genotyped via the Agena Biosciences (formerly Sequenom) iPLEX® Gold MassArray platform. Multiplex primer extension was performed, and extension products were analyzed by MALDI-TOF mass spectrometry (*Tang et al., 2004*).

Single nucleotide polymorphism genotyping efficiency was set to greater than or equal to 0.90. The hypervariable region SNP MT16189 and SNP MT9540 did not meet this threshold and were dropped from the analysis. We also removed SNP MT9540 from the classification algorithm as the primers lacked specificity, which is consistent with the amplification of nuclear insertions of mitochondrial origin (NumtS) common in the human genome (*Hazkani-Covo, Zeller & Martin, 2010*). The final list of custom panel SNPs used to classify mitochondrial haplogroups along with their genotyping assay designs is given in Tables S1 and S2, respectively.

Using the genotype data we generated from the custom SNP panel, we employed Hi-MC and Haplogrep2 to determine mitochondrial haplogroup classifications in the PAGE I Study reference HapMap samples. We then compared the Hi-MC mitochondrial haplogroup classifications to the HapMap-reported classifications for Phase I samples (CEU, YRI, CHB/JPT) (*International HapMap Consortium, 2005*). We also compared Hi-MC haplogroup classifications to Haplogrep2-based haplogroup classifications for both Phase I and Phase III HapMap (MXL) samples genotyped using the custom SNP panel. For each Hi-MC-assigned haplogroup, we calculated percent concordance compared with HapMap-reported and Haplogrep2-assigned haplogroups. If the haplogroup was assigned

"unclassified" using Hi-MC, we removed the sample from the concordance calculation. Classifications were considered concordant if they were in the same haplogroup, even if one classification method resulted in finer resolution. For example, if one method classified a sample as A2 and another method classified the same sample as A2x, such classifications were considered concordant. Differences in the resolution of haplogroup classifications were not unexpected given differences in underlying methodology and the number of SNPs used for classification. The HapMap-reported classifications were generated using more mitochondrial SNP genotypes compared to the reduced number of SNPs necessary to use Hi-MC. HapMap Phase I complete data includes genotypes for 214 mitochondrial SNPs, 49 of which overlap with the custom SNP panel genotyped in this study (Table S3). Additionally, Hi-MC uses a reduced tree for classification while Haplogrep2 employs all of Phylotree which can result in finer sub-haplogroup resolution.

As a final comparison, we accessed the 1,000 Genomes Project mitochondrial data for overlapping DNA samples with the PAGE I Study reference HapMap samples (45 CEU, 49 YRI, 85 CHB/JPT, and 60 MXL). The 1,000 Genomes Project data were generated by whole genome sequencing (*The 1,000 Genomes Project Consortium, 2010*, *2012*) and offer 3,892 sequence-based variants for mitochondrial haplogroup classification (*Rishishwar & Jordan, 2017*). We applied Haplogrep2 to these 1,000 Genomes Project data and compared haplogroup assignments with Hi-MC assigned haplogroups based on the custom SNP panel (63 SNPs) and HapMap-reported haplogroups (based on 214 SNPs).

## RESULTS

Here we compared HapMap-reported haplogroups (from ftp://ftp.ncbi.nlm.nih.gov/hapmap/mtDNA_and_chrY_haplogroups/) to haplogroups assigned by Hi-MC and Haplogrep2 (*Weissensteiner et al., 2016*) using a custom SNP panel we genotyped in HapMap DNA samples as part of the PAGE I Study (*Crawford et al., 2015*; *Matise et al., 2011*). We also performed similar comparisons for Haplogrep2-assigned haplogroups using 1,000 Genomes Project (*The 1,000 Genomes Project Consortium, 2010*, *2012*) mitochondrial SNP data for overlapping HapMap DNA samples. Overall, concordance between Hi-MC with both HapMap-reported (94.4%) and Haplogrep2-assigned (93.3%) haplogroups was high for CEU (Table 1A). Concordance in CEU was highest in the comparison between Haplogrep2 assignments and HapMap-reported haplogroups (97.8%; Table 1A). All five discordant haplogroup calls between Hi-MC and Haplogrep2 assignments were a result of missing genotypes at key SNPs for the Hi-MC classification tree (Table S4; Fig. S1). We observed two discordant haplogroups when comparing Haplogrep2 assignments with HapMap-reported haplogroups. One was observed in CEU sample NA12146, which has the MT11251 and MT15452 genotypes that support the JT Haplogrep2 haplogroup assignment (Table S4). Another was for CEU sample NA12875 where Haplogrep2 assigned JT while both HapMap reported and Hi-MC assigned H despite the sample having the JT-defining MT11251 genotype (Table S4). In general, the comparisons made with Haplogrep2 using 1,000 Genomes Project data supported high concordance for all CEU comparisons (Table 1B). Neither NA12146 nor NA12875 was included in the 1,000 Genomes Project.

**Table 1 Percent concordance in CEU and YRI populations for pair-wise comparisons of mitochondrial haplogroup classifications.**

**(A)**

|  | CEU ($n$ = 90) (%) | YRI ($n$ = 76)[a] (%) | CHB/JPT ($n$ = 85)[b] (%) |
|---|---|---|---|
| Hi-MC-assigned vs HapMap-reported | 94.4 | 90.8 | 45.9 |
| Hi-MC-assigned vs Haplogrep2-assigned | 93.3 | 59.2 | 84.9 |
| Haplogrep2-assigned vs HapMap-reported | 97.8 | 60.5 | 40.0 |

**(B)**

|  | CEU ($n$ = 45) (%) | YRI ($n$ = 45)[c] (%) | CHB/JPT ($n$ = 80)[d] (%) |
|---|---|---|---|
| Hi-MC assigned vs HapMap-reported | 93.3 | 91.1 | 40.0 |
| Hi-MC assigned vs Haplogrep2-assigned | 93.3 | 95.6 | 95.0 |
| Haplogrep2-assigned vs HapMap-reported | 100 | 95.6 | 41.3 |

Notes:
We assigned haplogroups using Hi-MC and Haplogroup2 based on (**A**) genotype data in CEU, YRI, and CHB/JPT HapMap samples generated by a custom SNP panel targeting 63 mitochondrial SNPs. We compared these haplogroup assignments with HapMap-reported haplogroups available at ftp://ftp.ncbi.nlm.nih.gov/hapmap/mtDNA_and_chrY_haplogroups/. (**B**) We also performed the same comparisons using Haplogrep2 on mitochondrial data available on overlapping HapMap samples from the 1,000 Genomes Project. HapMap MXL samples are not included here as no HapMap-reported haplogroups are available for comparison.
[a] Fourteen YRI samples were "unclassified" by Hi-MC due to missing SNPs and were removed from these comparisons.
[b] Four CHB/JPT samples were "unclassified" by Hi-MC due to missing SNPs and one CHB/JPT sample was labeled "unknown" as the HapMap-reported haplogroup. All five were removed from these comparisons.
[c] Four YRI samples were "unclassified" by Hi-MC due to missing SNPs and were removed from these comparisons.
[d] Four CHB/JPT samples were "unclassified" by Hi-MC due to missing SNPs and one CHB/JPT sample was labeled "unknown" as the HapMap-reported haplogroup. All five were removed from these comparisons.

Among YRI, 14 HapMap samples were assigned "unclassified" by Hi-MC due to missing SNPs or missing key SNPs at the beginning of the classification tree (Table S5). Of the remaining 76 YRI samples for this comparison, 69 Hi-MC haplogroup assignments were concordant with HapMap-reported haplogroups (90.8%; Table 1A). In comparison, concordance was lower for Haplogrep2-assigned haplogroups when compared either with HapMap-reported (60.5%) or Hi-MC-assigned (59.2%) haplogroups (Table 1A). In contrast, the concordance was uniformly high across YRI comparisons when haplogroup assignments were made with Haplogrep2 using 1,000 Genomes Project data (Table 1A).

Examination of the YRI discordant and concordant haplogroup comparisons yielded several observations (Table S5). First, the discordance observed between HapMap-reported and both Hi-MC and Haplogrep2-assigned haplogroups was not supported by the genotype data. That is, HapMap-reported L1 haplogroups had genotypes supportive of L0 in the custom SNP dataset (such as MT9042T, MT9347G, and MT5442C; Table S5) as well as in the 1,000 Genomes Project dataset suggesting L0 is the correct haplogroup. Second, HapMap-reported and Hi-MC-assigned L3 haplogroups, including L3b, L3d, and L3e, were assigned N or Y haplogroups by Haplogrep2 when applied to the custom SNP genotype data despite the lack of N haplogroup-supporting genotypes (MT10398A in the Hi-MC classification tree) in these samples. The discordance was resolved when Haplogrep2 was applied to 1,000 Genomes Project data suggesting that L3 is the correct assignment for these samples (Table 1B; Table S5). The remaining discordant assignments were the result of a key missing genotype (MT10115) defining L2, resulting in misclassifications by Hi-MC compared with Haplogrep2 (Table S5; Fig. S2).

Compared with the CEU and YRI populations, we observed less concordance among the CHB/JPT samples particularly for the Hi-MC- and Haplogrep2-assigned comparisons

with HapMap-reported haplogroups (Table 1A). One HapMap-reported classification was listed as "unknown" (for NA19012), and four CHB/JPT samples were unclassified due to missing SNPs at key genotypes for Hi-MC. Of the remaining 85 CHB/JPT samples, there was high agreement (84.9%) between Hi-MC- and Haplogrep2-assigned haplogroups (Table 1A). The low concordance with HapMap-reported haplogroups was mainly the result of haplogroups reported in HapMap as B, C, D, or E despite the lack of genotype support for these haplogroups (Table S6; Fig. S3). This high concordance between Hi-MC and Haplogrep2 assigned haplogroups using both the custom SNP panel data (Table 1A) and the 1,000 Genomes Project data (Table 1B) suggest that the majority of haplogroups reported by HapMap (in ftp://ftp.ncbi.nlm.nih.gov/hapmap/mtDNA_and_chrY_haplogroups/) for CHB/JPT are incorrect.

The mitochondrial haplogroups for the samples of Mexican ancestry from HapMap Phase III (MXL) have not been previously reported. Samples in this dataset include 30 trios of Mexican ancestry from Los Angeles, CA. We applied Hi-MC to determine mitochondrial haplogroups in these samples and characterized the distribution of mitochondrial haplogroups among the MXL. Due to matrilineal inheritance of mtDNA, offspring have the same mitochondrial haplogroup as their mother; therefore, offspring were excluded when calculating the frequency distribution of mitochondrial haplogroups. We also excluded three samples with unclassified haplogroups and two samples with likely incorrectly assigned haplogroups due to missing genotypes, for a total of 55 unrelated MXL samples. Overall in the MXL samples, 89.1% of mitochondrial haplogroups identified were of Native American ancestry and 10.9% were of European ancestry (Fig. 2). The distribution of haplogroups in the HapMap MXL samples is similar to the distribution of haplogroups observed in Mexican Americans ascertained for the NHANES (*Mitchell et al., 2014b*).

To further evaluate the performance of Hi-MC, we compared the Hi-MC mitochondrial haplogroup classifications of MXL samples to Haplogrep2-assigned classifications applied to either the custom SNP panel or the 1,000 Genomes Project data. Among the 90 MXL samples, six were unclassified by Hi-MC due to missing data at multiple SNPs. Of the 84 classified MXL samples, there was high agreement between Hi-MC-assigned haplogroups and haplogroups assigned by Haplogrep2 based on the custom SNP panel (97.6%) and the 1,000 Genomes Project data (96.6%). Two MXL samples were misclassified by Hi-MC as haplogroup L3, and both of these samples were missing 11 key SNPs for the Hi-MC classification tree.

## DISCUSSION

Using a custom panel of mitochondrial SNPs that we previously applied to participants in the NHANES datasets (*Mitchell et al., 2014b*), we developed Hi-MC, a method for high-throughput classification of European, African, and Native American mitochondrial haplogroup lineages. We evaluated the performance of Hi-MC, and with genotype data from the custom SNP panel, demonstrate that Hi-MC performs comparably to the widely-used tool Haplogrep2. While Haplogrep2 is an excellent tool for mitochondrial haplogroup classification that accepts either sequence or SNP genotype data, it was

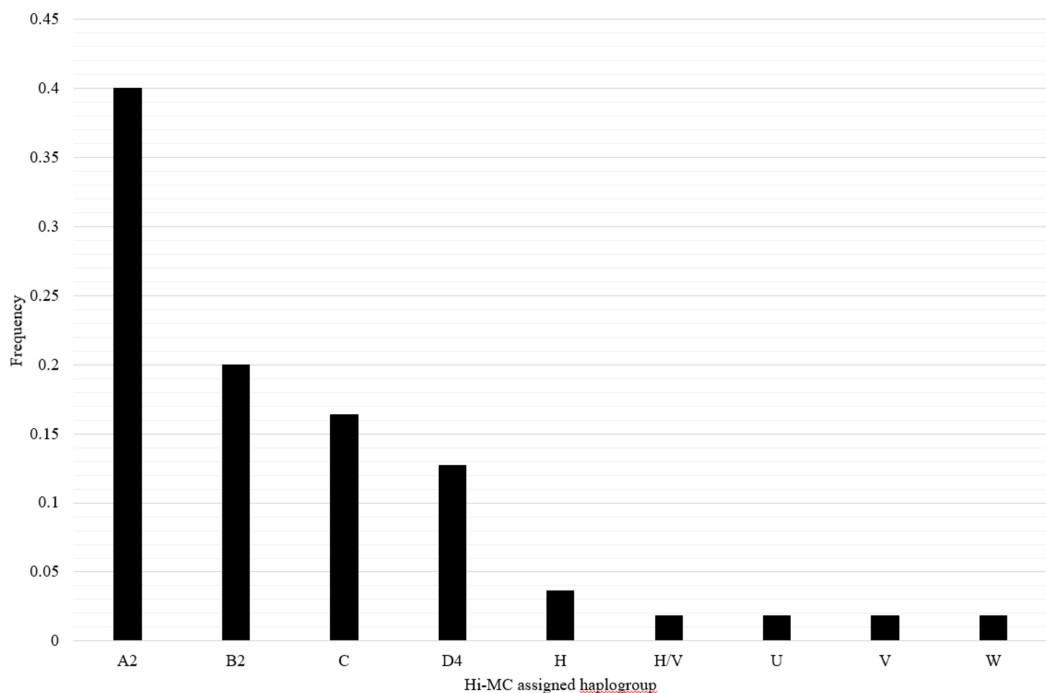

**Figure 2 Distribution of mitochondrial haplogroups in the HapMapMap Phase III samples of Mexican ancestry in Los Angeles, CA.** Hi-MC-assigned haplogroups based on the custom SNP panel for MXL samples and their frequency are plotted on the *X*- and *Y*-axes, respectively. We genotyped the custom SNP panel and applied Hi-MC to all MXL samples from Phase III of the HapMap Project ($n = 90$). Given that the mitochondrial haplogroup of the offspring is the same as that of the mother, offspring were excluded when determining the frequency distribution of haplogroups ($n = 30$). We further excluded three samples with unclassified haplogroups and two samples with likely mis-classified haplogroups by Hi-MC due to key missing SNPs. A total of 55 MXL unrelated samples are represented here.                         

developed primarily for sequence level data. The ability to alternatively genotype a relatively small number of SNPs allows for rapid haplogroup classification in a large number of genetic samples.

Mitochondrial SNPs captured by standard genome-wide genotyping arrays vary widely, and often the SNPs on these arrays are not informative for haplogroup determination. Hi-MC uses a defined panel of mitochondrial SNPs for classification of mitochondrial haplogroups eliminating the need for time-consuming SNP selection. Additionally, the relatively small number of SNPs in the custom panel makes Hi-MC particularly useful for large data sets where full mitochondrial genome sequencing is cost-prohibitive. As examples, approaches like Hi-MC promise to be of use to large biobank and cohort efforts such as the Million Veteran Program (*Gaziano et al., 2016*) and the UK Biobank (*Sudlow et al., 2015*), both of which continue to rely on cost-effective array-based assays rather than cost-prohibitive sequencing to generate genome-wide and mitochondrial data on hundreds of thousands to a million participants.

Hi-MC employs the commonly used PED/MAP file format as the input. There are a number of software programs that make use of the PED/MAP format, including PLINK (*Purcell et al., 2007*) which is widely used for analyzing genotypic data. Thus, in contrast to
Haplogrep, many Hi-MC users will not have to reformat data prior to use. Additionally, Hi-MC is an R-based software package that can be downloaded and run locally allowing for memory limits that are dependent on the machine where R is being run, which grants greater flexibility in the number of samples that can be processed at once. Once samples have been classified using Hi-MC, figures or tables displaying haplogroup frequencies can be easily generated via other R packages such as ggplot2 (*Wickham, 2009*).

We determined that Hi-MC performs well with samples of European, African, and Native American descent. Even though the Hi-MC custom SNP panel was not designed to specifically resolve Asian-specific haplogroups, we present evidence that Hi-MC is comparable to Haplogrep2 in haplogroup assignments for samples of Asian maternal lineage. Compared with other maternal lineages, the Asian branches of the mitochondrial phylogenetic tree are not as well-defined as other parts of the tree despite recent progress made in characterizing the phylogeny of Asian mtDNA (*Kivisild et al., 2002*; *Kong et al., 2006*). Another known challenge is that a major Asian haplogroup (B) is defined primarily by a deletion (8,281–8,289), a mutation not targeted by genotyping arrays. As more mtDNA sequences are obtained from individuals of Asian descent the phylogeny of mitochondrial genetic variation will be better understood. Future versions of Hi-MC will be updated to incorporate additional knowledge regarding subjects of Asian descent.

We applied Hi-MC to the HapMap Phase III MXL samples as the mitochondrial haplogroups for these participants have not been previously reported. The haplogroup distribution observed in the HapMap Phase III MXL samples is somewhat similar to the recently reported Haplogrep2-generated distribution for the MXL samples sequenced as part of the 1,000 Genomes Project (*Rishishwar & Jordan, 2017*). In this newer reference dataset, the most common reported haplogroup is A (25%) followed by B (15%) and C (9%) (*Rishishwar & Jordan, 2017*) compared with a higher A (A2) frequency in the present study (40%; Fig. 2). Overall, the distribution of Native American and European haplogroups in the MXL samples from HapMap Phase III is similar to the distribution observed in the NHANES Mexican American samples (*Mitchell et al., 2014b*). No African lineage mitochondrial haplogroups were identified among the HapMap MXL samples. This differs from the NHANES Mexican Americans in which 4.4% had mitochondrial haplogroups of African ancestry (*Mitchell et al., 2014b*). The lack of African haplogroups in the HapMap MXL samples is likely due to the small sample size and the regional ascertainment of these samples. While the NHANES samples were collected from across the United States, the HapMap Phase III MXL samples were ascertained solely from Los Angeles, CA, therefore are likely to be more homogeneous.

While there are several benefits to Hi-MC, there are some limitations. Currently, Hi-MC employs a reduced mitochondrial phylogenetic tree for classification. As a result, it is currently limited to classification of the major haplogroups of European, African, and Native American lineages, and requires that SNPs from the described custom panel be genotyped. While this panel was customized for populations expected for the PAGE I study, it is notable that several SNPs in this panel (MT1736, MT2092, MT3552, MT4883, MT10400, MT11177, MT11251, MT11719, MT12007, MT12308, MT12705, MT13368, MT14766) overlap with previously published panels (*Chaitanya et al., 2014*;

*Van Oven, Vermeulen & Kayser, 2011*), suggesting the potential for both greater resolution and generalizability in future extensions of Hi-MC. Additionally, because the method relies on a limited number of SNPs, it is not very robust to missing genotype data. And, while Hi-MC has the ability to classify mitochondrial haplogroups at a broad level, it currently cannot capture sub-haplogroups at finer resolution. As such, in instances where sequence level data are available another method for mitochondrial haplogroup classification, such as Haplogrep2, would be more appropriate.

Despite these limitations, Hi-MC offers several advantages including a defined panel of mitochondrial SNPs that is used in conjunction with the software for mitochondrial haplogroup classification. Hi-MC utilizes PED/MAP files for a user-friendly input file format, saving time and reducing opportunities for errors to be incorporated into the data. Also, Hi-MC is implemented in the commonly used statistical software environment R allowing for classification of relatively large sample sizes, as well as the ability to easily utilize other available R packages for visualization of results.

## CONCLUSIONS

We have developed a custom SNP panel and algorithm for mitochondrial haplogroup classification. The algorithm, Hi-MC, is implemented in R and makes use of PED/MAP file format for data input. We evaluated the performance of Hi-MC and demonstrate that classifications are comparable to the widely-used tool Haplogrep2. Hi-MC offers an algorithm that leverages a validated mtDNA SNP panel for mitochondrial haplogroup classification and is particularly valuable for studies in which sequencing is not feasible.

## ACKNOWLEDGEMENTS

Special thanks to Paxton Baker, MS, Melissa Allen, Ping Mayo, MS, and Nathalie Schnetz-Boutaud, PhD for their work in genotyping these samples. We also thank Dr. William Bush for his thoughtful comments.

### Funding

This work was supported by a National Institutes of Health grant (U01 HG004798) and associated American Recovery and Reinvestment Act (ARRA) supplements and institutional funds from the Institute for Computational Biology at Case Western Reserve University. There was no additional external funding received for this study. The funders had no role in study design, data collection and analysis, decision to publish, or preparation of the manuscript.

### Grant Disclosures

The following grant information was disclosed by the authors:
National Institutes of Health: U01 HG004798.
American Recovery and Reinvestment Act (ARRA).
Institute for Computational Biology at Case Western Reserve University.

## Competing Interests

The authors declare that they have no competing interests.

## Author Contributions

- Sandra Smieszek performed the experiments, analyzed the data, contributed reagents/materials/analysis tools, prepared figures and/or tables, authored or reviewed drafts of the paper, approved the final draft.
- Sabrina L. Mitchell conceived and designed the experiments, performed the experiments, analyzed the data, prepared figures and/or tables, authored or reviewed drafts of the paper, approved the final draft.
- Eric H. Farber-Eger performed the experiments, analyzed the data, contributed reagents/materials/analysis tools, prepared figures and/or tables, authored or reviewed drafts of the paper, approved the final draft.
- Olivia J. Veatch performed the experiments, analyzed the data, authored or reviewed drafts of the paper, approved the final draft.
- Nicholas R. Wheeler analyzed the data, contributed reagents/materials/analysis tools, authored or reviewed drafts of the paper, approved the final draft.
- Robert J. Goodloe performed the experiments, analyzed the data, contributed reagents/materials/analysis tools, authored or reviewed drafts of the paper, approved the final draft.
- Quinn S. Wells contributed reagents/materials/analysis tools, authored or reviewed drafts of the paper, approved the final draft.
- Deborah G. Murdock conceived and designed the experiments, contributed reagents/materials/analysis tools, authored or reviewed drafts of the paper, approved the final draft.
- Dana C. Crawford conceived and designed the experiments, contributed reagents/materials/analysis tools, prepared figures and/or tables, authored or reviewed drafts of the paper, approved the final draft.

## Data Availability

GitHub: https://github.com/vserch/himc.

Hi-MC (High-throughput Mitochondrial Haplogroup Classification): http://www.icompbio.net/resources/software-and-downloads/.

## Supplemental Information

Supplemental information for this article can be found online at http://dx.doi.org/10.7717/peerj.5149#supplemental-information.

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
