# Peer review of "Hi-MC: a novel method for high-throughput mitochondrial haplogroup classification"

_PeerJ, doi:10.7717/peerj.5149_

## Round 0.1 · original submission · Major Revisions

The reviewer suggestions look addressable, and we look forward to your revision.

Reviewer 1 ·

Basic reporting

The authors should provide a more detailed statistical basis for their haplogroup selection criteria. Why certain SNP combinations are better representatives in the minimalistic tree. Also the authors should provide some statistical analyses of genetic variation within each population, for example between ethnic groups.

In large number of mtDNA haplogroup studies many mtDNAs in a data set can not be assigned to the recognized haplogroups. To identify new basal lineages whole mtDNA sequencing is a must. How does Hi-MC handle this problem which can confound haplogroup assigments.

The authors should elaborate their rank calculation method mathematically. Without the exact details its difficult to assess the exact working of the algorithm. For example how does the algorithm incorporate the phylogenetic weight of each polymorphism in a particular combination of SNPs.

Experimental design

The experimental design is basically the same as Mitchell et al. Hum Genet. 2014. As mentioned above, without the detailed mathematical presentation of the algorithm its difficult to assess the exact working of the algorithm.

Validity of the findings

The conclusions are valid and the current limitations are well documented but the authors need to put some effort in explaining the basis of these limitations. Why are many Asian-specific haplogroups not well captured? What are the features that need to be improved for the next iteration so that the algorithm will perform well in most populations including Asian ancestry?

Additional comments

The authors should include a section detailing specific advantages and disadvantages of the different mitochondrial haplogroup classification methods.

mtDNA sequencing using NGS methods allow detection of mitochondrial DNA heteroplasmy with minor allele frequencies. The authors should include a section on how a selective SNP set can handle mitochondrial DNA heteroplasmy.

Could NuMTs play a bigger confounding role in a selective SNP panel based method given that any NuMT will have a higher mutation accumulation rate than a standard mitochondirail DNA and without the proper sequence context can complicate the results/inferences.

·

Basic reporting

The manuscript is well written, and a pleasure to read, with appropriate background and detail of the problem.

The manuscript suggests (line 110) that more information is available at http://www.icompbio.net. However, that website does not include information on this software. I clicked on several links but was not able to find a link to Hi-MC (a direct link to the page should be provided).

The readme file is helpful. However, an example dataset should be provided to enable readers to quickly test the tool before using it on their own data. For example, it was much easier to test drive haplogrep2 than Hi-MC because of the available testdata (https://haplogrep.uibk.ac.at). The release of source code is appreciated (unlike haplogrep2), along with MtoolBox and Phy-mer etc.

Authors are comparing with Haplogrep (Human Mutation 2011), though a newer method (haplogrep2) was released in July 2016, including an updated webserver.

Experimental design

It would be encouraging to see the R packing on CRAN/Bioconductor instead of GitHub. It provides confidence regarding the stability of the code, and enables better version control for reproducibility (especially Bioconductor).

Validity of the findings

As previous reviewers have pointed out; comparisons with other methods are critical to motivate readers to adopt this new tool. Along with Hapmap, 1000g data from phase 1, 3 and Li et al, can be used to benchmark the tool vs haplogrep, haplogrep 2 and Phy-Mer etc

Further, use of a custom Phylotree (a version of figure 1 Mitchell et al) can be tested with haplogrep 2 (standalone version), to compare the two tools using the same reference.

Although results are described well in the text, the accuracy of the findings at different resolutions can be conveyed in a graphical form. For example, figure 1 from Mitchell et. al can be used as a template and nodes can be used to display the number of correctly and incorrectly classified individuals per node (totaling haplogroups below the node). This should be done for Hi-MC and Haplogrep 2, and a supplementary table of the results of these two methods can be used to drive the discussion.

---

## Round 0.2 · accepted · Accept

We are pleased to accept your revised manuscript for publication.

# Reviewer 1 ·

Basic reporting

The manuscript is clearly written and meets the standard for publication.

Experimental design

Experimental design in the revised version is much more transparent than the original form.

Validity of the findings

The authors have addressed all the major queries comprehensively and this has led to a much improved manuscript.